# Clinical and Functional Characterization of a Novel URAT1 Dysfunctional Variant in a Pediatric Patient with Renal Hypouricemia

**Blanka Stiburkova** [1,2,*], **Jana Bohata** [1,3], **Iveta Minarikova** [1], **Andrea Mancikova** [4], **Jiri Vavra** [4], **Vladimír Krylov** [4] **and Zdenek Doležel** [5]

1   Institute of Rheumatology, Na Slupi 4, 128 50 Prague, Czech Republic
2   Department of Pediatrics and Adolescent Medicine, First Faculty of Medicine, Charles University and General University Hospital in Prague, Ke Karlovu 2, 120 00 Prague, Czech Republic
3   Department of Rheumatology, First Faculty of Medicine, Charles University, Na Slupi 4, 128 50 Prague, Czech Republic
4   Department of Cell Biology, Faculty of Science, Charles University, Vinicna 7, 128 00 Prague, Czech Republic
5   Department of Pediatrics, University Hospital Brno, Medical Faculty of Masaryk University, Jihlavska 20, 625 00 Brno, Czech Republic
*   Correspondence: stiburkova@revma.cz; Tel.: (+420)-234-075-319; Fax: (+420)-224-914-451

**Abstract:** Renal hypouricemia (RHUC) is caused by an inherited defect in the main (reabsorptive) renal urate transporters, URAT1 and GLUT9. RHUC is characterized by decreased concentrations of serum uric acid and an increase in its excretion fraction. Patients suffer from hypouricemia, hyperuricosuria, urolithiasis, and even acute kidney injury. We report the clinical, biochemical, and genetic findings of a pediatric patient with hypouricemia. Sequencing analysis of the coding region of *SLC22A12* and *SLC2A9* and a functional study of a novel RHUC1 variant in the *Xenopus* expression system were performed. The proband showed persistent hypouricemia (67–70 μmol/L; ref. range 120–360 μmol/L) and hyperuricosuria (24–34%; ref. range 7.3 ± 1.3%). The sequencing analysis identified common non-synonymous allelic variants c.73G > A, c.844G > A, c.1049C > T in the *SLC2A9* gene and rare variants c.973C > T, c.1300C > T in the *SLC22A12* gene. Functional characterization of the novel RHUC associated c.973C > T (p. R325W) variant showed significantly decreased urate uptake, an irregular URAT1 signal on the plasma membrane, and reduced cytoplasmic staining. RHUC is an underdiagnosed disorder and unexplained hypouricemia warrants detailed metabolic and genetic investigations. A greater awareness of URAT1 and GLUT9 deficiency by primary care physicians, nephrologists, and urologists is crucial for identifying the disorder.

**Keywords:** *SLC22A12*; URAT1; hypouricemia; uric acid transporters; excretion fraction of uric acid

## 1. Introduction

Hypouricemia is defined as serum uric acid concentrations below 119 μmol/L (2 mg/dL). It is characterized by increased uric acid clearance or decreased uric acid production. Hypouricemia is a relatively rare condition, occurring in about 0.15–3.3% of the general population and 1.2–4% in hospitalized patients [1,2]. Malignancy is ranked first as a possible etiology of secondary hypouricemia, followed by diabetes mellitus, renal tubulopathies such as Fanconi syndrome, and medication. Excretion fraction of uric acid (EF-UA) is a key biochemical marker for a differential diagnosis of primary hypouricemia. Markedly elevated EF-UA suggests renal hypouricemia while lower or normal EF-UA suggests hereditary xanthinuria [3].

Renal hypouricemia (RHUC) is a heterogeneous hereditary disorder caused by a dysfunction of the main renal urate transporters, URAT1 and GLUT9. Characteristic biochemical markers include markedly decreased serum uric acid concentrations (S-UA) and elevated EF-UA. Clinical markers include exercise-induced acute renal failure, urolithiasis, and hematuria along with fatigue, nausea, vomiting, and diffuse abdominal discomfort. However, RHUC is also characterized by clinical variability, and only about 10% of all patients with a URAT1 defect have nephrolithiasis and/or acute kidney injury due to spasms of the renal artery. Currently there is no treatment for RHUC; however, allopurinol has been used to prevent recurrence of acute kidney injury episodes, and oral supplementation with antioxidants is recommended [4].

The role of URAT1 and the association of genetic variants of the *SLC22A12* gene with renal hypouricemia (RHUC type 1, OMIM no. 220150) were identified in 2002 [5], and to date, about 200 patients have been identified. The relationship between the GLUT9 transporter (gene *SLC2A9*) and renal hypouricemia (RHUC type 2, OMIM no. 612076) was reported in 2008 [6], and to date, about 15 patients have been identified. Homozygous or compound heterozygous loss-of-function mutations in the *SLC22A12* gene lead to a partial defect in absorption of uric acid, while variants in the *SLC2A9* are responsible for severe hypouricemia and hyperuricosuria (SUA < 10 µmol/L, EF-UA > 90%), which is often complicated by nephrolithiasis and acute kidney injury, such as that seen in RHUC1. Genetic variants in the *SLC22A12* gene are the primary cause of renal hypouricemia (>90%) with major variants reported in Asia region (Japanese and Korea, variant p.W258X with frequencies 2.3%) and in the Roma population (p.L415_G417del and p.T467M with frequencies of 1.9% and 5.6%, respectively) [7–10].

This case study expands our understanding of the molecular mechanisms of renal hypouricemia and confirms the distribution of dysfunctional URAT1 variants in non-Asian patients.

## 2. Materials and Methods

### 2.1. Patient

A three-year-old Caucasian girl was referred for an endocrine examination due to her small stature. The child's mother had been under long-term treatment for a psycho-affective disorder, which also included the pregnancy with her daughter; the father was healthy. The child was born during the 31st week of pregnancy, by C-section, due to premature discharge of amniotic fluid. Birth weight was 1330 g, and length was 37 cm. Oxygen therapy was necessary for 5 days but without the need for artificial pulmonary ventilation. During development, the child showed symptoms of psychomotor retardation. Therefore, developmental rehabilitation was initiated. Rehabilitation was carried out, however, family compliance was poor. The mother abandoned the family, and the girl was in alternating custody of her father and grandmother. On initial examination by an endocrinologist, the girl was 94.5 cm tall (−3.4 SD), had a body weight of 13.4 kg (−1.8 SD), and had only grown about 2.7 cm in the previous year. The father's body height was 163.5 cm, but the mother's height was unknown. The child's physical examination was without irregularities except for orbital hypertelorism. Her level psychomotor development corresponded approximately to that of a two-year-old child. Because hypouricemia (67–70 µmol/L) was repeatedly found during endocrine re-examinations, further analyses were carried out, mainly focused on purine metabolism disorders. High-performance liquid chromatography determination of hypoxanthine and xanthine in urine was performed on Waters Alliance 2695 [3].

### 2.2. Genetic Analysis

Genomic DNA was extracted from a blood sample using a QIAmp DNA Mini Kit (QiagenGmbH, Hilden, Germany). All coding exons and intron-exon boundaries of *SLC22A12* and *SLC2A9* were amplified from genomic DNA using polymerase chain reaction and subsequent purificated using a PCR DNA Fragments Extraction Kit (Geneaid, New Taipei City, Taiwan). DNA sequencing was performed on an automated 3130 Genetic Analyzer (Applied Biosystems Inc., Foster City, CA, USA). Primer

sequences and PCR conditions used for amplification were described previously [11,12]. The reference genomic sequence was defined as version NC_000011.8, region 64,114,688..64126396, NM_144585.3 for *SLC22A12*; NM_020041.2, NP-064425.2, SNP source dbSNP 132 for *SLC2A9*.

### 2.3. Functional Analysis

A missense variant of URAT1, p.R325W, was tested for urate transport activity using in vitro expression analysis in *Xenopus* oocytes as previously described [11,13]. Subcellular localization was determined using immunocytochemical analysis. Immunodetection of URAT1 was performed on 3.5 μm paraffin sections using rabbit anti-SLC22A12 polyclonal antibody (Sigma, St. Louis, MO, USA). The paraffin sections were stained after heat-induced antigen retrieval (10 mM citrate buffer, pH 6.1, for 20 min at 97.0 °C in a water base) using standard blocking procedures. The primary antibody against URAT1 was diluted 1:25 in PBS and applied overnight at 4 °C. Detection of bound primary antibodies was achieved using Alexa Fluor 488-conjugated with anti-rabbit IgG (diluted 1:500; Abcam, Cambridge, Britain). For image acquisition, we used an Olympus BX53 fluorescent microscope (Olympus, Hamburg, Germany).

## 3. Results

### 3.1. Patient

The proband had persistent hyperuricemia (67–70 μmol/L; ref. range 120–360 μmol/L) and an increased EF-UA (24.3–34.2%; ref. range 7.3 ± 1.3%) with normal urinary excretion of hypoxanthine and xanthine, Table 1. No clinical or laboratory symptoms of renal disease were present in the patient. During follow-up, the patient was without episodes of acute kidney injury; her ultrasound exam showed no nephrolithiasis, and her creatinine clearance (estimated as eGF) was within the normal range. Supporting examinations to determine the cause of her small stature found normal serum levels of Thyroid Stimulating Hormone (TSH) and fT4, however, IGFBP-3 was low, and IGF-1 was well below the reference level. A growth hormone deficit was demonstrated with stimulation tests (clonidine test, hypoglycemia test). Brain magnetic resonance imaging found a markedly small hypophysis; the pituitary stalk was evaluated as normal. Cytogenetic analysis demonstrated a normal 46, XX karyotype. Growth hormone replacement therapy was initiated.

**Table 1.** Biochemical and genetic parameters of the proband and her father. [a] reference range for women and children; [b] reference range for men.

| Table *Cont.* | Serum UA (μmol/L) | EF-UA (%) | Serum Creatinine (μmol/L) | Identified Variants in *SLC22A12* |
|---|---|---|---|---|
| **Proband** | 67–70 | 24.3–34.2 | 32 | c.973C > T (C/T); c.1300C > T (C/T) |
| **Father of proband** | 205 | N/A | 62 | c.1300C > T (C/T) |
| **Reference range** | 120–360 [a] 120–420 [b] | 7.3 ± 1.3 [a] 10.3 ± 4.2 [b] | 50–110 | - |

### 3.2. Genetic Analysis

Sequencing analysis of SLC2A9 revealed six intron variants (rs2240722, rs2240721, rs2240720, rs28592748, rs13115193 and rs61256984), three synonymous variants (rs13113918, rs10939650 and rs13125646) and three common non-synonymous allelic variants (rs2276961, p.G25R; rs16890979, p.V282I and rs2280205, p.P350L). A sequencing analysis of SLC22A12 revealed one intron variant (rs11231837), four synonymous (rs3825016, rs11231825, rs1630320, and rs7932775), and two heterozygous rare non-synonymous allelic c.973C > T variants (p.R325W, rs150255373, Figure 1A,B), and a previously identified c.1300C > T variant (p.R434C, rs145200251) [14]. Segregation analysis was not fully performed because the child's family was not interested.

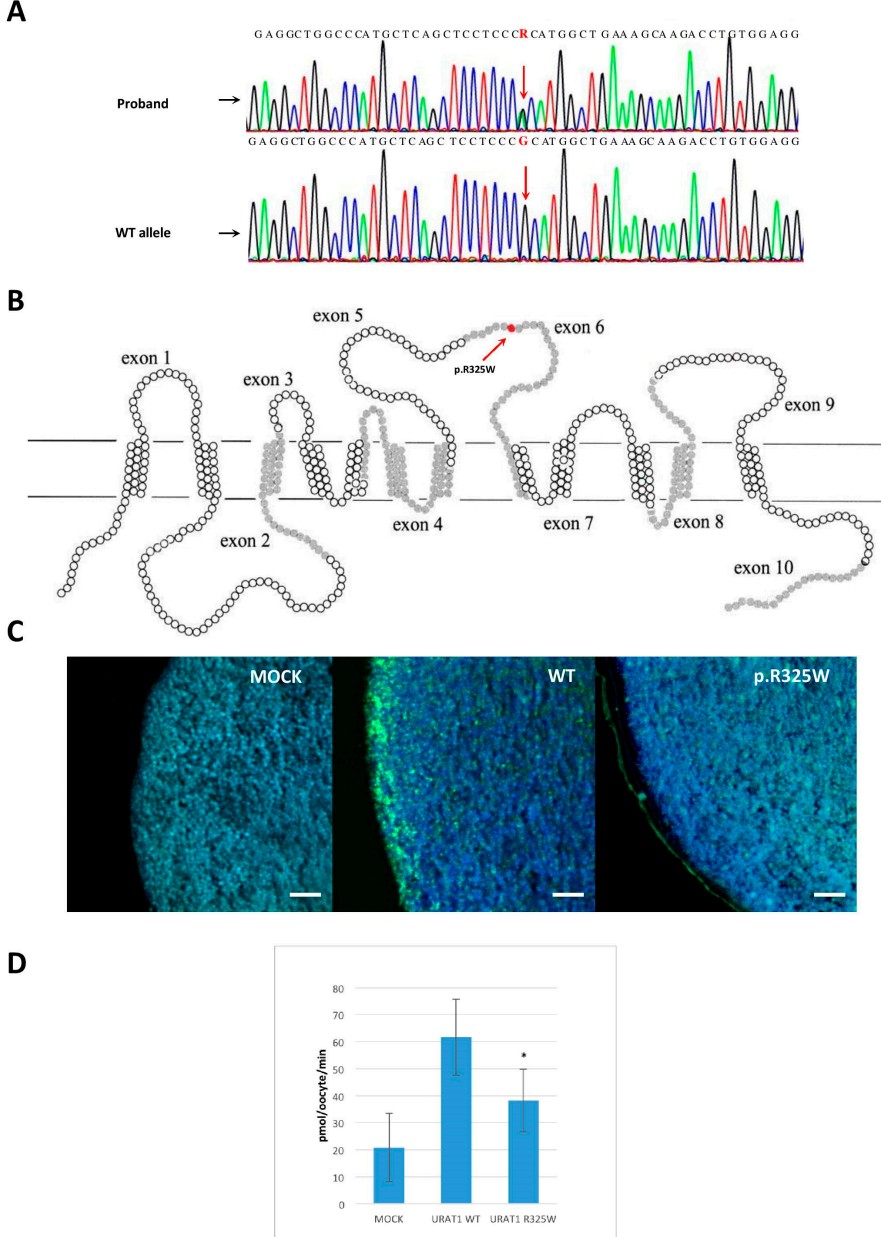

**Figure 1.** Illustration of allelic variant p.R325W in genetic, protein and functional context. (**A**) Electropherograms of partial sequences of exon 6 showing a heterozygous c.973C > T variant in the *SLC22A12* gene. (**B**) Position of identified allelic variant p.R325W in a URAT1 membrane topology model. (**C**) Immunohistochemical analysis of *Xenopus* oocytes injected with 50 ng of cRNA encoding the wt or p.R325W using anti-URAT1 polyclonal antibodies. The URAT1 signal is green, autofluorescent granules in the cytoplasm of oocytes are blue. Water-injected oocyte without any detectable URAT1 signal. Oocyte injected with wt cRNA exhibited a strong linear signal on the plasma membrane and a finely granular intracytoplasmic signal. The variant p.R325W was characterized by a weak discontinuous URAT1 signal on the plasma membrane, and reduced intracytoplasmic staining compared to the wt. Scale bar represents 50 μm. (**D**) Uric acid accumulation in *Xenopus* oocytes transfected with wt URAT1 and p.R325W URAT1 allelic variant after 30 min of incubation in [8-$^{14}$C] 600 μM uric acid/ND - 96 solution. The data was tested by One-way Analysis of variance (ANOVA). In comparison to wt URAT1, significantly lower UA accumulation (* $p < 0.05$) was detected in p.R325W URAT1 injected oocytes (n = 5; means ± SD). $H_2O$ injected (mock) oocytes were used as a negative control.

*3.3. Functional Analysis*

Urate transport via the p.R325W variant was significantly decreased in comparison to the wild type (wt) (* $p < 0.05$), Figure 1C. This finding indicated that the above-mentioned URAT1 variant leads to reduced urate reabsorption at the apical membrane of proximal renal tubules leading to decreased serum urate levels. Oocytes expressing the wt exhibited strong continuous URAT1 immunostaining on the plasma membrane and dispersed finely granular staining in the cytoplasm. Oocytes expressing p.R325W showed a weak discontinuous URAT1 signal on the plasma membrane and intracytoplasmic staining was lower than in the wt, Figure 1D.

## 4. Discussion and Conclusions

Urate transport in the kidney is a complex process involving several transmembrane proteins that provide reabsorption (URAT1, GLUT9) and secretion (ABCG2). In genome-wide association studies, the *SLC2A9* gene is a well-established locus that is significantly associated with hyperuricemia while the *ABCG2* locus had the most significant association with gout susceptibility [15–17]. The dysfunctional variants in URAT1 and GLUT9 cause hereditary renal hypouricemia, and genetic analysis is needed to confirm the diagnosis and/or to identify the specific type of renal hypouricemia.

The *SLC22A12* gene is located on chromosome 11q13. Ten exons encode two transcript variants of the URAT1 transporter (332 and 553 amino acids), which are specifically expressed on the epithelial cells of the proximal tubules in the renal cortex [4]. At present, 52 variations in the *SLC22A12* coding region (40 missense/nonsense, two splicing, three regulatory, three small deletions, two small insertions, one gross deletion, and one complex rearrangement) have been described (HGMD Professional 2018.4, http://www.hgmd.cf.ac.uk). Thirty-six URAT1 variants are currently associated with the hypouricemia phenotype. Functional analysis confirmed in part of these variants impact on urate uptake ability and/or cytoplasmatic expression and localization [7,11,13,14]. However, not all URAT1 allelic variants have effect on decreasing of protein expression and/or function (p.R228E, R477H) [7,11].

The analysis of *SLC2A9* coding regions in our proband revealed three common non-synonymous variants: heterozygous rs2276961 (p.G25R, Caucasian MAF = 0.53), rs16890979 (p.V282I, Caucasian MAF = 0.21), and homozygous rs2280205 (p.P350L, Caucasian MAF = 0.48). These variants have not been previously reported in association with hypouricemia. Variant p.V282I was previously described relative to the hyperuricemia and gout phenotype [18]. Moreover, in our previous study, which used association analysis together with functional and immunohistochemical characterization of these variants identified in the adult population, we did not find any influence of these allelic variants on expression, subcellular localization, or urate uptake of GLUT9 transporters [19].

Our analysis of *SLC22A12* coding regions revealed two rare heterozygous non-synonymous variants: rs150255373 (p.R325W, Caucasian MAF = 0.001) and rs145200251 (p.R434C, Caucasian MAF unknown). Variant p.R434C was previously associated with renal hypouricemia 1 in a five-year-old Macedonian girl suffering from distal renal tubular acidosis and renal hypouricemia [14]. The patient had symptoms of dehydration, polyuria, and vomiting. The patient also had rickets and slow growth. There was evidence of hyperchloremic metabolic acidosis (pH 7.23, HCO$_3$ 13.6 mmol/L, BE = 12.6 mmol/L), hypokalemia (3.0 mmol/L), hypophosphatemia (0.84 mmol/L), hypouricemia (73 μmol/L), and hyperuricosuria (EF-UA 24–31%). Bilateral nephrocalcinosis and a solitary cyst in the left kidney were discovered during an ultrasound examination. The patient was given alkali therapy; metabolic compensation was achieved, serum electrolytes normalized, and low molecular proteinuria resolved. Only the hypouricemia parameter persisted during the two-year observational period. The mother of this patient was a heterozygote for the same missense variant (S-UA 136 μmol/L, EF-UA 19%) and a history of renal colic and the passage of a single renal calculus. Functional studies of p.R434Cs were previously performed using transiently transfected HEK293 cells [14]. Plasma membrane expression levels of the p.R434C variant were low, intracellular localization was not strongly observed, and urate uptake showed a significant reduction of urate transport function (P < 0.001).

The structural model for URAT1 is mainly focused on the organization and alignment of residues within 12 transmembrane spanning domains. Variant p.R325W was localized within the putative extracellular loop. This variant has not yet been identified in the patients of those with renal hypouricemia, but the nature of this mutation strongly suggests that it is pathogenic; PolyPhen software (http://genetics.bwh.harvard.edu/pph/) suggested that the variant is possibly damaging (score 0.72). Moreover, another predictive software, SIFT (http://sift.jcvi.org/), suggests that this variant is deleterious (score 0). On the other hand, CADD (https://cadd.gs.washington.edu/), REVEL, and MetaLR predictive software indicate that the impact of the variant is likely to be tolerated or benign. In the middle stands Mutation Assessor (http://mutationassessor.org/r3/) which predicts a moderate functional impact. Evolutionary analysis of URAT1 paralogs, including six mammalian species, revealed conservation of p.R325W only between human and chimpanzee (Figure 2). Human and Simian monkeys possess high affinity and low capacity URAT1 transporter which diverged from the original low affinity, high capacity paralog (mouse, rat, horse and dog) 43 MYA [20]. Authors described four key amino acid substitutions in human URAT1 positions 25, 27, 365 and 414 as a crucial for the high to low affinity and low to high capacity shift. Similarly, as a variant p.R325W, all four amino acid residues are conserved between human and baboon, but not among other mammalian species. The functional characterization of p.R325W showed significantly decreased urate uptake and a weak, discontinuous URAT1 signal on the plasma membrane and reduced intracytoplasmic staining. The results suggested that p.R325W variant may not affect URAT1 function qualitatively (via alteration of its intrinsic transporter activity), but rather do so quantitatively (via decreasing its cellular protein level). Taken together, the data confirm the causality of the p.R325W variant relative to renal hypouricemia 1.

**Figure 2.** Alignment of the p.R325W URAT1 amino acids in the studied allelic variants with chimpanzee, horse, dog, rat and mouse paralogs.

Detailed investigations of serum uric acid concentrations and excretion fractions of uric acid in patients with unexplained hypouricemia are needed. Many patients with RHUC may be asymptomatic; however, pediatric nephrologists know that RHUC can cause acute renal failure, especially after strenuous physical activity. Another risk of RHUC is the development of nephrolithiasis. Although renal hypouricemia is a rare hereditary disorder, the frequency of novel URAT1 associated variants shows that this condition is underdiagnosed. RHUC should be considered not only in patients from Japan or Asia. The phenotypic severity of RHUC1 is not correlated with results from functional characterizations of URAT1 variants. Functional studies regarding the impact of novel associated variants are necessary to determine their correlation with scores from prediction algorithms and to confirm causality.

**Author Contributions:** Conceptualization, B.S.; validation, B.S.; J.B. conducted sequencing analyses; I.M., A.M., J.V. and V.K. worked on experiments using *Xenopus* oocytes, and analyzed the data; Z.D. was responsible for clinical observations; data curation, B.S.; writing, B.S.; project administration, B.S.; funding acquisition, B.S.

**Funding:** Supported by the Ministry of Health of the Czech Republic: AZV 15-26693A, the project for conceptual development of research organization 00023728 (Institute of Rheumatology) and RVO VFN64165.

**Conflicts of Interest:** The authors declare no conflict of interest.

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
