# Peer review of "Clinical and Functional Characterization of a Novel URAT1 Dysfunctional Variant in a Pediatric Patient with Renal Hypouricemia"

_applsci, doi:10.3390/app9173479_

Round 1

Reviewer 1 Report

In hypouricemic patient, mutation in SLC22A12 gene, which encodes apical urate transporter URAT1, was found and its effect on URAT1 activity/expression was conducted. Mutations were also found in SLC2A9, which encodes basolateral urate transporter GLUT9. However, they are known not to be associated with hypouricemia. Accordingly, SLC22A12 mutation was considered responsible for hypouricemia. Then, c.973C>T mutation was newly found to cause decreased expression and activity by in vitro experiments, the mutation was suggested to be associated with hypouricemia. This study is clear and mostly acceptable. There are following comments.

1: URAT1 mutation in R325W showed lowered expression of protein in membrane surface of Xenopus oocytes. In addition, apparent uptake of urate by the mutant is lower than wild type. Here, it is not clear whether the decreased apparent activity was due to reduced protein expression or function per URAT1 protein. As described in Discussion, transport activity can be evaluated by affinity and capacity of urate via URAT1. So, experimental demonstration of affinity of the mutant R325W in comparison with wild-type should be scientifically important, since this genetic disease is rare but significant.

2: line 19,

--- in the main renal (reabsorptive) transporters. Insert, (reabsorptive) to make it clearer.

3: line 151, [9-15],

What does this number mean?

4: Figure D, H20, 2 should be subscript.

Author Response

1: URAT1 mutation in R325W showed lowered expression of protein in membrane surface of Xenopus oocytes. In addition, apparent uptake of urate by the mutant is lower than wild type. Here, it is not clear whether the decreased apparent activity was due to reduced protein expression or function per URAT1 protein. As described in Discussion, transport activity can be evaluated by affinity and capacity of urate via URAT1. So, experimental demonstration of affinity of the mutant R325W in comparison with wild-type should be scientifically important, since this genetic disease is rare but significant.

Thank you for your comments. According to this advice, we revised the text in Discussion.

The results suggested that p.R325W variant may not affected URAT1 function qualitatively (via alteration of its intrinsic transporter activity), but rather do so quantitatively (via decreasing its cellular protein level).

2: line 19,

--- in the main renal (reabsorptive) transporters. Insert, (reabsorptive) to make it clearer.

Corrected according to suggestion.

3: line 151, [9-15], What does this number mean?
Corrected according to suggestion. [9-15] is typo only, it should be [8-14C].

4: Figure D, H20, 2 should be subscript.
Corrected according to suggestion.

Reviewer 2 Report

Review of the manuscript applsci-562292

The authors report on a case of hypouricemia with mutations in main uric acid transporters, namely URAT1 and GLUT9. They structurally and functionally analysed the new URAT1 variant R325W showing besides other mutations possibly affecting the reabsorption of uric acid in the kidney in this patient it significantly reduced URAT1-mediated transport. I really enjoyed the analyses of the variants and their possible impact. The methods are sound and the results support the discussion. Reports like these are very important as they are showing which variants matter in patients revealing the bigger picture of uric acid homeostasis in patients. However, I have some comments the authors should consider:

1.       It would be nice to have a table with the patient parameters tested as well as the detected mutations.

2.       How did the authors measure hypoxanthine and xanthine? Could you please add this to the method part?

3.       In Figure 1D, could the authors please add statistical analysis data for H2O vs URAT1wt? And to keep it consistent with figure 1C the authors should call ‘H2O’ in figure 1D ‘mock’ as well. The figure legend needs proper proofreading! I guess the pR375W is just a typo and should be R325W?

4.       Some minor comments:

-          The official short form for the unit ‘litre’ is capital ‘L’. Please correct this for ALL unit descriptions!!

-          Lines 89 – 91: please carefully check the wording here!

-          Line 98: “A new missense variant of URAT1, …“, please add!

-          Lines 152 – 153: it is H2O and you should mention that this is considered mock!

-          Lines 154 – 155: do not split figure legends, please!  

-          Line 157: “Urate transport in the kidney is ….”, please add!

Author Response

It would be nice to have a table with the patient parameters tested as well as the detected mutations.
Corrected according to suggestion, we have added the Table.
How did the authors measure hypoxanthine and xanthine? Could you please add this to the method part?
Corrected according to suggestion, methodology was added at the end of 2.1 Patient part. High-performance liquid chromatography determination of hypoxanthine and xanthine in urine was performed on Waters Alliance 2695 [3].

In Figure 1D, could the authors please add statistical analysis data for H2O vs URAT1wt? And to keep it consistent with figure 1C the authors should call ‘H2O’ in figure 1D ‘mock’ as well. The figure legend needs proper proofreading! I guess the pR375W is just a typo and should be R325W?
Corrected according to suggestion, statistical analysis was added. (One-way ANOVA). H20 was corrected in figure 1D to mock Some minor comments:
-          The official short form for the unit ‘litre’ is capital ‘L’. Please correct this for ALL unit descriptions!!
Corrected according to suggestion in all descriptions.

-          Lines 89 – 91: please carefully check the wording here!
Corrected according to suggestion. All coding exons and intron-exon boundaries of SLC22A12 and SLC2A9 were amplified from genomic DNA using polymerase chain reaction  and subsequent purificated using a PCR DNA Fragments Extraction Kit (Geneaid, Taiwan).

-          Line 98: “A new missense variant of URAT1, …“, please add!
Corrected according to suggestion.

-          Lines 152 – 153: it is H2O and you should mention that this is considered mock!
Corrected according to suggestion.

-          Lines 154 – 155: do not split figure legends, please!
Corrected according to suggestion.

-          Line 157: “Urate transport in the kidney is ….”, please add!
Corrected according to suggestion.

Reviewer 3 Report

The authors identified a variant form of URAT1 (R325W) in a three-year-old Caucasian girl with dysfuntional uric acid metabolic status and renal hypouricemia. Using Xenopus oocytes injection assay, they demonstrated that the R325W mutant URAT1 displayed decreased urate uptake, altered cellular localization and a dampened URAT1 signal. I recommend publication of this manuscript after a few clarifications.

1. In Figure 1D, I suggest to add uric acid accumulation data of another known mutant (R477H or R228E) which has been reported to show not significant difference from WT.

These mutants were reported in Reference11. Eur J Jum Genet 2013, 21, 1067 and Kidney Int. 2004, 66, 935

2. A scale bar should be added in Figure 1C to show the size of the cells. Also, it is confusing what the white arrow is.

3. The authors should add an overall conclusion sentence before figure legend in Figure 1, splitting Figure 1 into multiple figures if needed.

4. Grammar and typos should be corrected. For example:

Page3 line 130: Segregation analysis was not performed

Page4 line 142: p. R325W 

page1 line 41: Excretion fraction of uric acid (FE-UA) should be EF-UA according to the authors' writing.
Other places that are not listed here should also be corrected.

Author Response

In Figure 1D, I suggest to add uric acid accumulation data of another known mutant (R477H or R228E) which has been reported to show not significant difference from WT.

Thank you for your suggestion. According to this comment, we added these information in Discussion since we think is not methodologically right to combine data from more studies into one graph. We would really appreciate it if our decision could be respected.

 Functional analysis confirmed in part of these variants impact on urate uptake ability and/or cytoplasmatic expression and localization [7,11,13,14]. However no all URAT1 allelic variants have effect on decreasing of protein expression and/or function (p.R228E, R477H) [7,11].

These mutants were reported in Reference11. Eur J Jum Genet 2013, 21, 1067 and Kidney Int.2004, 66, 935

A scale bar should be added in Figure 1C to show the size of the cells. Also, it is confusing what the white arrow is.
Corrected according to suggestion. Scale bar was added and white arrow was removed. The authors should add an overall conclusion sentence before figure legend in Figure 1, splitting Figure 1 into multiple figures if needed.
Conclusion sentence was added: illustration of allelic variant p.R325W in genetic, protein and functional context. Grammar and typos should be corrected. For example:

Page3 line 130: Segregation analysis was not performed
Corrected according to suggestion.

Page4 line 142: p. R325W 
Corrected according to suggestion.

page1 line 41: Excretion fraction of uric acid (FE-UA) should be EF-UA according to the authors' writing.
Other places that are not listed here should also be corrected.
Corrected according to suggestion in whole article.

Round 2

Reviewer 1 Report

Authors responded well and revised manuscript well. There is no more comment on this manuscript.

Reviewer 3 Report

The authors have addressed my concerns.